# Renaissance of VDAC: New Insights on a Protein Family at the Interface between Mitochondria and Cytosol

**DOI:** 10.3390/biom11010107

**Published:** 2021-01-15

**Authors:** Vito De Pinto

**Affiliations:** 1Department of Biomedicine and Biotechnology Sciences, University of Catania, Via S. Sofia 64, 95123 Catania, Italy; vdpbiofa@unict.it; Tel.: +39-095-73842444; 2we.MitoBiotech.srl, c.so Italia 172, 95129 Catania, Italy; 3National Institute of Biostructures and Biosystems, Section of Catania, 00136 Rome, Italy

**Keywords:** Voltage-Dependent Anion selective Channel, isoforms, oxidative post-translational modification, gene promoter, yeast, bioenergetics, metabolism

## Abstract

It has become impossible to review all the existing literature on Voltage-Dependent Anion selective Channel (VDAC) in a single article. A real Renaissance of studies brings this protein to the center of decisive knowledge both for cell physiology and therapeutic application. This review, after highlighting the similarities between the cellular context and the study methods of the solute carriers present in the inner membrane and VDAC in the outer membrane of the mitochondria, will focus on the isoforms of VDAC and their biochemical characteristics. In particular, the possible reasons for their evolutionary onset will be discussed. The variations in their post-translational modifications and the differences between the regulatory regions of their genes, probably the key to understanding the current presence of these genes, will be described. Finally, the situation in the higher eukaryotes will be compared to that of yeast, a unicellular eukaryote, where there is only one active isoform and the role of VDAC in energy metabolism is better understood.

## 1. Introduction

The study of Voltage-Dependent Anion selective Channel (VDAC), at that time more commonly called mitochondrial porin, broke by chance in the laboratory of Prof. Palmieri, at the University of Bari, where I was an internal student and then researcher. The goal of the laboratory was to isolate and characterize the mitochondrial carriers, today grouped in the family of solute carriers (SLC25) to which Prof. Palmieri has given a decisive contribution [1]. As it was later understood, the mitochondrial porin has physical-chemical characteristics very similar to those of the SLC25 family, since it is a protein deeply immersed in the phospholipid membrane with very few portions exposed to the aqueous solvent. For this reason, VDAC was initially thought to be another contaminant solute carrier obtained during the purification procedures of the phosphate transporter which was the primary target of the laboratory [2]. VDAC, although found in the outer membrane, shared with the SLC25 family present in the inner membrane, a very similar molecular weight (around 30 kDa) and a similar affinity for the stationary chromatographic phase of hydroxyapatite in purification procedures [3,4]. This made it very difficult to distinguish VDAC from other integral membrane proteins. A big step forward was the use of radioactive dicyclohexylcarbodiimide (DCCD). This ATP-synthase inhibitor, at very low concentrations, was able to mark only three proteins in mitochondria: an 8 kDa protein, the c subunit of ATPase, a band of about 16 kDa and one of about 35 kDa. Excluding the c subunit band, so hydrophobic to be soluble in apolar solvents, the 35 kDa DCCD-binding protein was observed in our laboratory as one of the bands in the crowded Mr 30–35 kDa area in SDS-PAGE. The DCCD binding was used as a specific indication for the purification of what was considered one of the putative carriers [2]. It was named DCCD-binding protein, pending the discovery of a functional activity [5]. After numerous attempts to identify any specific substrate exchange activity with the techniques used by us, an intuition by Prof. Palmieri led me to the laboratory of Prof. Roland Benz, then at the University of Konstanz, where the purified protein (personally carried by hand in a large Dewar jar) showed a powerful and immediate pore-forming activity in planar artificial membranes [5,6]. The presence of a mitochondrial pore-forming protein in *Paramecium* extracts was first claimed in 1976 [7] and then the functional identification of this protein as a component of the mitochondrial outer membrane was first reported by Colombini in 1979 [8]. The study of this pore-forming protein distinguished this research from the more established one of the laboratory, which was related to transport proteins of the inner mitochondrial membrane. Nevertheless, the technologies, then in full development, for the study of integral membrane proteins could be applied to both types of proteins. For example, a modification of the chromatography with hydroxyapatite and celite allowed the production of large amounts of VDAC, with a very simple methodology that eventually became standard in all laboratories in the world [9]. With this methodology various structural approaches were attempted but resulted as only partially successful. Following a course of crystallization of membrane proteins held in Martinsried (DE) with teachers as Hartmut Michel and Johann Deisenhofer (who were awarded the Nobel Prize for photosynthetic reaction center the following year), a large preparation of VDAC in Triton X-100 was thrown away, as the detergent prevented the formation of crystals. It was in that context that more modern and dialyzable detergents were tested for the first time on VDAC purification, such as LDAO [10]. The use of this detergent, which was later adopted by all laboratories, led to the crystallization of VDAC, which was obtained a good twenty years later [11,12,13]. Another structural aspect that, in retrospect, can be considered one of the most important was the identification of the VDAC DCCD-binding amino acid residue. DCCD binds negatively charged residues exposed to a hydrophobic environment: a physical-chemical apparent incongruity that made the binding of DCCD to mitochondrial membrane proteins so rare. The identification procedure advanced with a direct but artisanal experimental strategy, given the instrumental means available at the time. In the end, however, VDAC1 bovine heart glutamate 73 was identified as the binding site of DCCD [14]. The transmembrane arrangement of the protein and its secondary structure was not yet known, although there were predictions based not only on bioinformatics [15,16] but mostly on the electrophysiology data obtained by Colombini’s and Forte’s groups [17,18]. VDAC is the hexokinase-binding protein on the outer membrane surface [19] and binding of hexokinase to VDAC was found to be inhibited by DCCD [20]. In [20] it was proposed that the C-terminal end of VDAC contained the DCCD binding site. The identification of E73 as the DCCD-binding residue did not fit with the contemporary models, since the residue was initially located in an outside loop of the current folding pattern [17,18]. The crystal structure of the pore definitely solved the dispute, locating the E73 in the middle of a transmembrane β-strand, facing the hydrophobic phospholipid layer [11,12,13]. The hexokinase-VDAC binding has intriguing functional implication that continues to be highly relevant and whose mechanism has not yet been clarified.

## 2. The Next Twenty Years of Achievements 

Studies focused on the biochemical-structural aspects of the protein underwent a strong acceleration following the use of molecular biology technologies that became within the reach of all laboratories. In the case of VDAC, this led to a great expansion of knowledge of the protein’s genetic and cellular activities.

The milestones in a twenty years path of achievements, in my opinion, were: (I) the identification of three isoforms of VDAC in the superior metazoa [21,22]. (II) The definition of the structure of VDAC1, obtained in the same year by three different groups with different techniques (crystallization and NMR) [11,12,13], and of VDAC2 [23]. Surprisingly, the structure proposed a new type of domain: a mixed β-barrel with odd number of β -strands (19 β-strands), i.e., with the presence of parallel β-strands (the first and the last) in addition to the antiparallel strands. (III) The topological arrangement of VDAC in the outer membrane [24]. (IV) The functional discovery of the oligomerization of VDAC and its role in hexokinase binding and apoptosis triggering [25,26,27,28]. (V) The involvement of VDAC in many pathologies, from tumors [29] to neurodegenerative diseases like ALS [30], Parkinson’s disease [31], Alzheimer’s disease [32], type 2 diabetes [33], and the identification of it as a potential therapeutic target [34,35,36,37].

The discovery of more isoforms of VDAC suggests that evolution developed variants with slight amino acidic differences in its protein armory for precise but still undefined purposes. The production of stem cells and mice knockout for the individual isoforms has allowed us to start providing clues about their function [38,39,40,41]. A major part of our current research endeavor resides in this question, i.e., in identification of the function of individual isoforms.

## 3. VDAC Isoforms: A Puzzle to Unveil

Most of the literature about VDAC1 was covered in the excellent and extensive review by Shoshan-Barmatz in this special issue [42]. We will thus focus on two aspects that, in our opinion, deserve further attention: the presence in the genome of more VDAC isoforms and their utilization.

In higher metazoa three VDAC genes with the same exon-intron structure [43] evolved: while the nucleotide changes among the three VDAC isoforms modified the encoding sequences so that they have peculiar differences (such as the cysteine content, which will be discussed later), they have not affected the structure of the splicing sequences nor have they modified the gene organization. The only exceptions are for the VDAC2 gene, which appears identical to the other two but with the addition of an extra exon upstream of the first one, which gives VDAC2 a short additional sequence to the N-terminal; and for VDAC3, where the presence of an internal starting codon (ATG), resulting in the insertion of a single methionine residue at amino acid position 39 of the mature VDAC3 protein, was reported but whose relevance was not established [44,45]. Notably, the function of VDAC2 is still unknown. Descending the evolutionary scale, for example, the additional exon of VDAC2 is no longer present in fish [23]. The relevance of the individual isoforms of VDAC has been addressed by the development of knockout cells for the individual isoforms [39,40]. Surprisingly, the overall proclaimed result was that each VDAC isoform, individually, is not needed for cell survival [38]. This notion was especially obtained to exclude the presence of VDAC in the permeability transition pore structure (PTP) in which it was previously involved [41]. On the other hand, the non-essentiality of the existence of VDAC strongly clashes with the abundance with which nature has provided the mitochondrion with this protein.

## 4. The Most Abundant Post-Translational Modifications of Mammalian VDACs Occur on Cysteines

We have been studying post-translational modifications of VDAC isoforms for some years now. The initial starting point was the consideration of the different number of cysteines present in VDAC isoforms and the suspicion that this difference was not a mere coincidence but was likely linked to a specific function or structural involvement of these residues in protein folding (Figure 1). In particular, since the VDAC3 isoform was the least studied, a set of mutagenesis of the individual cysteines and/or of small clusters, and even all cysteines in the sequence, was undertaken. These mutants were tested for their electrophysiological activity after in vitro expression, purification and reconstitution in a planar lipid bilayer, and in yeast devoided of VDAC pore-forming activity following endogenous gene inactivation [44]. The results clearly showed an inverse correlation between the number of cysteines present in the VDAC3 sequence and the reconstituted pore-forming activity. Additionally, the recovery of the fermentation activity of the mutant yeast progressed when it was transformed with VDAC3 in which the cysteines were progressively eliminated [46,47]. At the same time, work by another group [48] proposed that the cause of the reduced activity of VDAC3 when cysteines were present in its sequence was due to the formation of intra-chain disulfide bridges [49].

The need to investigate this functional result more thoroughly led us to collaborate with the mass spectrometry unit to unequivocally highlight, at a molecular level, the oxidation state of cysteines, in particular, and of other residues in the VDAC3 isoform, first [50], and in other isoforms later [51,52]. While some of these experiments are still ongoing, especially the part regarding the presence of intra-chain disulfide bridges, the results of our studies are summarized in the following sections.

### 4.1. Methods: Avoiding Unreliable Results

The technique of protein isolation from animal cells and/or tissues has been modified and adapted to eliminate any risk of oxidation or accidental modification due to isolation protocols or electrophoretic techniques. We used enriched extracts rather than purifying proteins after SDS-PAGE or 2D electrophoresis. To this end, reduction and alkylation of sulfur were performed on purified mitochondria, and only later the chromatographic separation was run. The eluted proteins were cleaned by PlusOne 2-D Clean-Up kit (GE Healthcare Life Sciences, Milan, Italy), then RapiGest SF (Waters, Milan, Italy), to eliminate non-protein contaminating molecules. The sample was then subjected to proteolytic cleavage and the peptide mixture loaded on UP-nanoLC and then analysed by a highly sensitive Orbitrap Fusion Tribrid (Q-OT-qIT) mass spectrometer. This modified procedure is all the more delicate and important to develop when considering that the main PTM studied was the oxidation of -SH [53,54].

### 4.2. Cysteine Oxidations in VDAC Isoforms

We initially focused on oxidative post-translational modifications of VDAC3 cysteines [46,50] and later on other isoforms [51,52]. In addition, starting materials from different organisms such as rat tissues and human cell cultures were compared. The first novel finding was that VDAC cysteines can undergo progressive oxidation of sulphur. Some oxidations are reversible, others are difficult to reverse or are practically non-reversible under physiological conditions. The oxidation state of sulphur can go from the redox-reactive thiol (–SH) to sulfhydration (SSH), disulfide bonds (RS-SR), sulfenylation (SOH), sulfinic acid (SO_2_H), and sulfonic acid (SO_3_H) [55]. Except for sulfonic acid, all the reported oxidative post-translational modifications are readily reversible. The variable oxidation of cysteines could be due to the presence of ROS, particularly abundant in the intermembrane space, where there is also an acidic pH favourable to oxidation. The question we asked was whether these modifications were random or precisely targeted. In fact, VDAC, despite being an integral membrane protein, has the inner side of the pore and the loops exposed to the water environment: the walls of the hydrophilic channel and the connection loops between the β-strands are exposed to the inside and outside of the outer mitochondrial membrane. Moreover, the location of the N-terminal segment (amino acids 1–19), which contains portions of α-helix, is not unequivocally defined and even less is known about the structure and location of the further distal segment at the N-terminal (11 additional amino acids) found in the VDAC2 sequence. Most of the cysteines of VDAC2 and VDAC3 indeed are exposed to the aqueous environment and in certain situations are close enough to each other to suggest that they may engage in the formation of disulphide bridges. Taken together, the finding of cysteine oxidative post-translational modifications [52,53,54,55] indicate that each VDAC sulphur amino acid has a preferential sensitivity to oxidation. Indeed, some cysteines oscillate between different oxidative states (from reduced to sulfinic acid), others are always irreversibly oxidized (sulfonic acid), while many others are always reduced (for a detailed review see [55]). Therefore, the latter cysteines have the potential to form disulfide bonds. The propensity to oxidation is a preserved characteristic of single cysteines depending on their location within the sequence and therefore in the 3D structure of the pore: those in the same position, in different organisms, have the same type of oxidation [55]. Moreover, this propensity to oxidation of cysteines is peculiar to VDAC because no other mitochondrial proteins isolated with the same chromatography technique show the presence of oxidized cysteines [51]. The significance of the oxidative modifications peculiar to VDAC could modulate an unknown function of the proteins, and/or the buffering of the oxidative potential of the ROS produced in the mitochondrion [52].

### 4.3. Other Post-Translational Modifications in VDACs

In VDACs other more common post-translational modifications were detected. For example, phosphorylation is very common and dynamic [56]. In our hands, in particular, Ser 104 was usually found phosphorylated, but, in general, in low amounts [55].

Acetylation was always detected at the N-terminal amino acid of the three isoforms [50,51], together with the loss of the starting methionine. Furthermore, succinated cysteines were not found in human VDAC1 isoform but were exclusively found in VDAC2 and 3 [51]. No selenocysteine was found, as well as no evidence of ubiquitin and ubiquitination was detected. A very rare and unique post-translational modification, the deamidation of specific asparagine and glutamine was found in cultured NSC34 cells transformed to express the SOD1G93A variant: this cell line is the most used cell model of ALS. It is tempting to speculate that this modification might be associated with the pathology [54].

## 5. VDAC Isoforms Genes Expression Regulation: The Key to Understanding Isoforms Functions?

The studies of the structure, activity, and regulation of VDAC genes aim at obtaining a reliable picture of their differences in tissue expression or sensitivity to specific stimuli [57]. In humans, for each VDAC gene, several different transcript splice variants were identified: they did not vary in the coding region but mainly in the length of their 5′-UTR and 3′-UTR. This finding led us to the hypothesis that there might be different mechanisms of transcript regulation and expression in various cellular contexts. Other splice variants were detected, including processed transcripts that do not contain open reading frame (ORF), retained intron, and transcripts involved in the nonsense-mediated decay mechanism. It is not known whether the identified VDACs splice variants have any functional biological role. However, gene expression data collected from NIH Genotype-Tissue Expression project (GTEx) [58], report their transcription, including the generation of non-protein coding transcripts. 

### 5.1. VDAC Genes Expression Profile

All three VDAC isoforms are ubiquitously expressed, with the highest levels found in skeletal and heart muscles as determined by RNA-seq GTEx. The level of the VDAC1 and VDAC2 transcripts is comparable, while VDAC3 is lower than the other two isoforms, confirming previous experimental conclusions drawn by RT-PCR [59]. While confirming that VDAC isoforms are ubiquitously expressed, the comparison with the data present in a second repository (RNA-seq CAGE RIKEN FANTOM 5’ project) [60] of Expression Atlas repository of EMBL-EBI [61] revealed that the VDAC3 expression levels were higher than VDAC2 and VDAC1, whose transcripts are scarcely represented in all tissues [57]. The data emerging from this analysis highlight for the first time the prevalence of VDAC3 gene transcription as compared to other isoforms reflecting a higher promoter activity. The special version of RNA-seq methodology based on cap analysis of gene expression adopted by the FANTOM5 consortium explains the difference in the results obtained in the former database.

### 5.2. VDACs Genes Promoter Structures and Activity

The promoters of the human VDACs genes were also characterized. The organization of the core promoter is similar to that of most TATA-less human promoters of ubiquitously expressed genes, where the presence of abundant GC regions, alternative binding sites Inr, DPE, and BRE assure a basal levels of transcription. Interestingly, a non-canonical initiation site termed the TCT motif (polypyrimidine initiator), which is a target for translation regulation by the mTOR pathway, oxidative, and metabolic stress [62], was also identified in VDAC2 and VDAC3 promoters [57]. Gene reporter assays revealed that VDAC3 promoter had the highest transcriptional activity and VDAC1 promoter was, on the contrary, the least active [57]. We proposed that a quantitative regulation of the transcript levels due to their different stability, or to maintain a high level of transcripts to promptly respond to a particular stimulation, is necessary for the cell.

### 5.3. Specific Functions of Transcription Factors Binding Sites in VDAC Genes Promoters

The basal expression level of VDAC genes seems to be subject to quantitative regulation of expression. Using a bioinformatics approach, the main transcription factors regulating the activity of VDAC promoter regions were identified, and a quest for the corresponding binding sites located in the promoters was performed. In all three VDACs promoters, the majority of identified TFs classes belong to the E2FF, NRF1, SP1, KLFS, EBOX families which are prevalently involved in cell proliferation and differentiation, apoptosis, and metabolism regulation [63,64,65,66,67,68,69]. This result suggests that VDAC expression may have a central role in regulating mitochondrial function. VDAC promoters are also equipped with unique transcription factor binding sites. These transcription factors may be the key to understanding the difference among the VDAC isoforms in terms of binding sites specific to the promoters of each VDAC gene.

The unique transcription regulators in VDAC1 promoter suggest a prevalent role of this protein in the mitochondrial outer membrane in physiological context and in altered environmental conditions in which cells have to restore the mitochondria energy balance [70,71,72]. VDAC2 promoter showed the presence of different factors specially active during the development of specialized tissues and organogenesis processes mainly related to nervous system genesis and growth [73,74]. VDAC3 promoter analyzed in [57] shows a particular abundance of transcription factors binding sites involved in the development of germinal tissues, organogenesis, and sex determination [39,75]. Converging evidence reported in the literature confirms the crucial role of VDAC isoforms in the specific context where the transcription factors that bind to their promoters exert a function.

In conclusion, the study of the features of the promoters of each VDAC isoform indicates that they evolved different control sequences, requiring transcription factors that link these genes to specific functions. Interestingly, the accumulated evidence points to the same biological area that was involved in the functions of the proteins.

The families of transcription regulators identified as unique in VDAC1 promoter suggest that this isoform has the main role of mitochondrial channel protein in a physiological cell context and is the main tool used to maintain mitochondria energy balance [40]. Several observations, obtained by experimental work, showed the involvement of VDAC1 in regulating cellular and mitochondrial pathways in physiology and pathology [76]. VDAC2 was indicated as the isoform indispensable for apoptosis [41,77,78] and autophagy in various cellular contexts [79]. VDAC2 promoter contains TF binding sites related to factors specially involved in developing specialized tissues, in particular of the nervous system, and the organogenesis and development processes. VDAC3 promoter is rich in GC repeats, which are typically found in epigenetic control systems of the expression of the transcript. Results from VDAC3 knockout mice show that the gene deletion affects the sperm organization and mobility [39]: the abundant presence of TF binding sites involved in germinal tissue development and sex determination might confirm its role in spermatogenesis.

## 6. The Role of VDAC Isoforms in Yeast

Unlike vertebrates, unicellular eukaryotic organisms such as *Neurospora crassa* and *Saccharomyces cerevisiae* do not have the same gene multiplicity for VDAC isoforms. The case of yeast has been studied and definitively clarified by us. After the discovery and characterization of yeast VDAC [80,81], Forte’s group, using genetic ablation of the gene coding the former VDAC protein, discovered a second isoform, named yVDAC2, with a partially conserved sequence, which was predicted to form channels [82]. Our group expressed the yeast VDAC2 isoform and characterized its activity after reconstitution in planar membranes. We found that recombinant yVDAC2 is also able to fold forming a pore and that its electrophysiological characteristics are very similar to those of yVDAC1 [83,84]. While the expression pattern and the physiological role of yVDAC2 remain unknown, knockout out of yVDAC1 forces cells to adopt a fermentative energy metabolism, highlighting the failure of yVDAC2 to compensate for the lack of yVDAC1 [82].

Proteomic analysis of the yeast mitochondrion, recently carried out at such high resolution as to allow a realistic estimate of the number of individual proteins present per mitochondrion [85], showed that yVDAC2 is present in an almost infinitesimal amount, even under conditions of stimulation of its presence [85]. The lack of support of a physiological role of yVDAC2 in a yVDAC1-free mitochondrion was also highlighted at the transcription level: a microarray experiment analyzing the transcriptomic profile of a yeast strain without VDAC1 (*Δpor1* mutant) grown on glucose showed that there is no increase in the expression levels of yVDAC2. Together, these results indicate that there is no coordination between the expression of the two genes; the fact that there is a second VDAC gene could be the result of a gene duplication that has led to the presence of some sort of inactive pseudogene in the yeast genome [86].

Even the analysis of the global permeability of the outer mitochondrial membrane of yeast, while highlighting the possible existence of other minor (quantitatively) proteins capable of forming pores, found no evidence of a yVDAC2 contribution to it [87]. We calculated that at least 90% of the permeability of the mitochondrial outer membrane is due to yVDAC1 and that the only proteins that can realistically support the permeability of small metabolites in the absence of yVDAC1 are the Tom40 and Sam50 channels (for a complete discussion see [86]).

The main unresolved questions are: how can the yeast mitochondria not be degraded and inactivated in the absence of VDAC1? How can the yeast mitochondria survive when VDAC1 is absent?

The transcriptomic profile of the *Δpor1* strain compared to the wild type (WT) strain provided interesting information. There is a marked aliquot of genes whose expression is completely modified in *Δpor1*: particularly impressive is the full inactivation of the mitochondrial genome. This drastic change can be attributed to the reduced transport capacity of nucleotides in the absence of VDAC1, also due to the impossibility to import carriers, a function recently linked to VDAC1 [88]. In these conditions, the mass of the organelle is reduced by more than 65% [86]. This result indicates a truly essential role of yVDAC1 in mitochondrial physiology. In fact, deletion of yVDAC1 causes a general reorganization of energy metabolism, as evidenced by a large number of up- and down-regulated genes associated with glycolysis, alcoholic fermentation, oxidative phosphorylation, TCA cycle, and lipid synthesis [86]. Our results showed that, in the absence of yVDAC1, cells survive by shifting the pyruvate metabolism from mitochondrion to cytosolic acetyl-CoA production by PDH bypass. This change leads to an increase in fatty acids and phospholipids that go into intracellular deposits or contribute to extending the size of the plasma membrane, as indicated by detailed microscopy experiments. Overall, these results indicate that VDAC1 in yeast contributes to the global regulation of energy metabolism.

## 7. Today’s Big Challenges—Conclusions

I hope that this quick review of the biological functions related to the onset of VDAC isoforms during evolution supports the hypothesis that they are involved in specific functions, as well as having, all of them, the ability to form large, porous aqueous channels. It is difficult to identify functional specificities based on small structural differences, such as the disposition and reactivity of some residues, the presence of small amino acid traits whose 3D organization and mobility have not yet been precisely defined, as it is the case of N-terminal extensions. These clues will have to be explored and revealed, together with the other major unsolved issue of this protein: the mechanism of voltage dependence. The road is open in this direction, and the end of the knowledge of the functions of VDAC isoforms will make it possible to steer the path of discovery of therapeutic molecules targeting this intriguing pore.

## Figures and Tables

**Figure 1 biomolecules-11-00107-f001:**
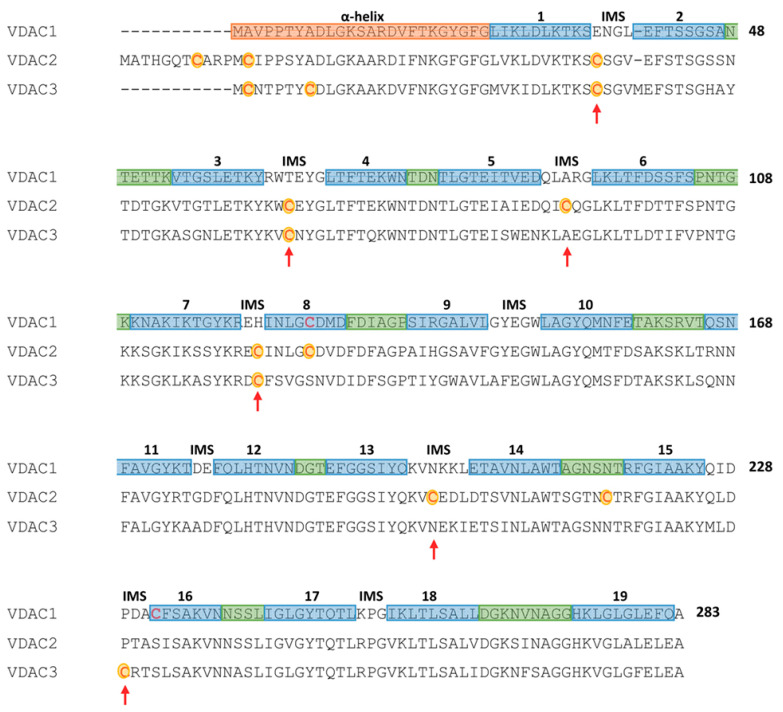
Secondary structure elements and cysteines localization in a multi-alignment of human Voltage-Dependent Anion selective Channel (VDAC) isoforms. Color code: light blue: β-strands; green: loops exposed to cytosol; orange: the N-terminal sequence containing α-helical portions; no color: loops exposed to inter-membrane space. Cysteine residues of VDAC2 and VDAC3 are in red and highlighted in yellow; red arrows point cysteine residues exposed to inter-membrane space (Figure obtained with the support of F. Zinghirino).

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
