# Peer review of "Renaissance of VDAC: New Insights on a Protein Family at the Interface between Mitochondria and Cytosol"

_biomolecules, 2021, doi:10.3390/biom11010107_

Round 1
Reviewer 1 Report
This review is a fairly comprehensive review of Voltage dependent Anion Channels, although the topic is quite extensive. The author describes some of the history of VDACs, especially those areas that the author participated in. I would recommend inclusion, however brief, of the work in paramecium and the 1979 Nature publication of Marco Colombini's first description of the mammalian outer membrane permeability pore. The author could also mention the single codon alternative splicing in VDAC3. Otherwise, the article would benefit from English language editing, there are a variety of grammatically awkward phrases, typos (eg line 216), and incorrect word choices (eg,line 67 “backwards”, the author likely means “in retrospect”). Some of the discussion is overly detailed, eg, enhanced protein purification, other areas a bit vague, eg, transcription factor binding sites.
Author Response
I am simply grateful to the Reviewer for the appreciation on our work. I have modified the text accordingly to the comments and inserted new references, as required. English text has been edited by a colleague living in the US since 15 years.
Reviewer 2 Report
This paper provides a comprehensive and informative review of up-to-date literature on VDAC isoforms genetics in cellular, organismic, and evolutionary contexts. This is an elegantly written, very timely review with interesting historical detours and a personal touch. This review would be appealing to a wide audience of cell biologists and would especially useful to young researchers. The Author appropriately starts his paper with “It has become impossible to review all the existing literature on VDAC in a single article”. Indeed, lately, there is a risk for multiple review papers on VDAC to get lost among other similar papers repeating information on VDAC properties and listing VDAC involvement in an array of different pathologies. The Author avoided this common path and found new territory by focusing on the latest data (mostly by his group) on gene expression and regulation of VDAC isoforms and on their post-translational modifications. The discussion on VDAC isoforms evolution is certainly sound. The part on the role of VDAC isoforms in yeast is informative – it provides a comparison with mammalian VDACs and summarizes the available literature on this subject. The Author, whose own contribution to the development of a widely used method of VDAC purification from mitochondria of different sources is well recognized, presents an interesting first-hand story of how VDAC research started in the early 80s. This provides an attractive personal touch to the paper.
The paper is well-written and provides an up-to-date review of the relevant works. However, it needs thorough proofreading. I have a few comments which could help readers clarify a few points.
- Line 73: “The transmembrane arrangement of the protein and its secondary structure was not yet known, although there were many bioinformatics predictions [13,14]”. In fact, the first VDAC folding pattern of 16 beta-strands was proposed by Song and Colombini (JBB, 1996), and was based not only on bioinformatic predictions but mostly on electrophysiology data obtained by Colombini’s and Forte’s groups. That should be acknowledged in this context. The folding pattern of VDACs from different species suggested by Marco Colombini was the most recognized and accepted VDAC “structure” before 2008 when its 3D structure was solved. It was the only available structure at that time. By following the Author’s historic standpoint on VDAC studies, it is worth mentioning that E73 residue was initially located in the outside loop in Colombini’s folding pattern, not in the protein-lipid interface according to 3D structure, which could be a source of the still ongoing controversy surrounding this residue.
- Line 75: “Later it was determined that E73 was involved in the binding of hexokinase to VDAC”. Nakashima, JBB,1989 [Ref. 15] showed that “the C-terminal end of VDAC is involved in binding to the N-terminal end of hexokinase”, particular residues 257-265 and 275-283. First of all, this was not “later”, because it was published in 1989 which is earlier than the Author’s work in 1993, [Ref. 12]. Second, there is no identification of E73 residue as a hexokinase binding site in [Ref. 15]. The same comment applies to the work by Linden et al, FEBS Lett 1982 [Ref. 16] where they suggest that mitochondrial porin is a hexokinase-binding protein without mentioning its identity, VDAC, or the binding residue.
- Line 89: “The functional discovery of the oligomerization of VDAC” – it should be mentioned in this context that VDAC’s natural potency to form oligomeric arrays was first demonstrated by Carmen Mannella’s group (Gui et al, J Struct Biol 1995) followed later by the work of Scheurung’s group (Goncalves et al J Mol Biol 2007) where “supramolecular assembly of VDAC” was demonstrated by AFM in native mitochondrial outer membranes.
- Line 93: identification of VDAC “as a convenient therapeutic target [28–31].” I would avoid using the term “convenient” here. VDAC is a desirable pharmacological target but it is far from a “convenient” one. The Author has convincingly described the rather frustrating situation with the difficulties of approaching VDAC as a pharmacological target in one of his own previous reviews (Reina and De Pinto, 2017).
- Line 268: “VDAC2 was indicated as the isoform indispensable for apoptosis [35,67]” – please add a reference to the latest important work on VDAC2 and Bax, Bak by Chin et al Nature Communications 2018 DOI: 10.1038/s41467-018-07309-4.
- Line 281: Needs clarification that this is the VDAC2 yeast isoform.
- Line 282: What did the author mean by “The result is that recombinant yVDAC2 is also able to fold back to form a pore.”? Please revise.
Author Response
I am genuinely grateful to this Reviewer for his/her supporting notes and for having grasped so neatly the essence of our efforts.
I have modified the text accordingly to the comments and inserted new references, as required.
English text has been edited by a colleague living in the US since 15 years.
Reviewer 3 Report
This is a interesting and useful topical review by a leading researcher on mitochondrial VDAC.
I have one minor comment: Although the language is generally clear, English usage is occasionally awkward or non-idiomatic.
Editorial assistance by a native speaker might be helpful at the discretion of the author and editors.
Overall, the manuscript was a pleasure to read.
Author Response
I am truly gratefull for the comment. I was assisted by a colleague living in the US since 15 years for editing the text.